# Bladder Cancer: Immunotherapy and Pelvic Lymph Node Dissection

**DOI:** 10.3390/vaccines12020150

**Published:** 2024-01-31

**Authors:** Zhongru Fan, Junpeng Deng, Yutao Wang, Xin Fan, Jianjun Xie

**Affiliations:** 1Department of Urology, The Affiliated Suzhou Hospital of Nanjing Medical University, Suzhou Municipal Hospital, Gusu School, Nanjing Medical University, Nanjing 211166, China; dr_fanzhongru@163.com (Z.F.); djunpeng@outlook.com (J.D.); 2Department of Urology, Peking Union Medical Hospital, Beijing 100005, China; ytwang96@cmu.edu.cn; 3Department of Radiology, The Second Affiliated Hospital of Dalian, Medical University, Dalian 116044, China

**Keywords:** bladder cancer, immunotherapy, pelvic lymph node dissection

## Abstract

Bladder cancer, a common malignancy of the urinary system, is routinely treated with radiation, chemotherapy, and surgical excision. However, these strategies have inherent limitations and may also result in various side effects. Immunotherapy has garnered considerable attention in recent years as a novel therapeutic approach. It harnesses and activates the patient’s immune system to recognize and eliminate cancer cells, which not only prolongs therapeutic efficacy but also minimizes the toxic side effects. Several immune checkpoint inhibitors and cancer vaccines have been developed for the treatment of bladder cancer. Whereas blocking immune checkpoints on the surface of tumor cells augments the effect of immune cells, immunization with tumor-specific antigens can elicit the production of anti-tumor immune effector cells. However, there are several challenges in applying immunotherapy against bladder cancer. For instance, the efficacy of immunotherapy varies considerably across individual patients, and only a small percentage of cancer patients are responsive. Therefore, it is crucial to identify biomarkers that can predict the efficacy of immunotherapy. Pelvic lymph nodes are routinely dissected from bladder cancer patients during surgical intervention in order to remove any metastatic tumor cells. However, some studies indicate that pelvic lymph node dissection may reduce the efficacy of immunotherapy by damaging the immune cells. Therefore, the decision to undertake pelvic lymph node removal should be incumbent on the clinical characteristics of individual patients. Thus, although immunotherapy has the advantages of lower toxic side effects and long-lasting efficacy, its application in bladder cancer still faces challenges, such as the lack of predictive biomarkers and the effects of pelvic lymph node dissection. Further research is needed to explore these issues in order to improve the efficacy of immunotherapy for bladder cancer.

## 1. Introduction

Bladder cancer is a common malignancy of the urinary system. According to the most current data of the World Health Organization’s International Agency for Research on Cancer (IARC) on the worldwide cancer burden, bladder cancer ranks 10th in terms of incidence rate, with approximately 570,000 new cases diagnosed in 2020 alone [1]. Pathologically, it can be classified as metastatic urothelial carcinoma (mUC), non-muscle-invasive bladder cancer (NMIBC), or muscle-invasive bladder cancer (MIBC) [2,3]. NMIBC accounts for 80% of bladder cancer cases and has a favorable prognosis, with 5-year survival rates higher than 85% (2) and a low risk of progression to invasive disease [4,5]. Despite the higher survival rate, 30–50% of NMIBC patients experience frequent recurrences, and 10–20% progress to MIBC [6].

Currently, the treatment approaches for bladder cancer include radiation, chemotherapy, surgery, and immunotherapy. In this review, we have explored the current research on immunotherapy for bladder cancer, and we examine the impact of pelvic lymph node dissection (PLND) on the therapeutic efficacy.

## 2. Current Status of Immunotherapy for Bladder Cancer

Tumor immunotherapy aims to mobilize the body’s natural immune defense mechanisms or stimulate specific anti-tumor responses using vaccines with tumor-specific antigens. Significant progress has been made in the field of tumor immunology in recent years, and it is an emerging treatment strategy for bladder cancer (Table 1).

**Immunotherapy using BCG vaccine:** The Bacillus Calmette–Guérin (BCG) vaccine consists of liver-attenuated *Mycobacterium bovis* and induces innate immunity and long-term memory response against tuberculosis [7,8]. In addition, BCG is used for postoperative Intravesical chemotherapy of bladder cancer based on local non-specific immunity [9,10]. The local immune response induced by BCG intravesical therapy can be summarized into three basic steps: (1) BCG particles attach to the urothelium and are engulfed by antigen-presenting cells (APCs), resulting in the secretion of inflammatory cytokines associated with BCG-induced cytotoxicity; (2) CD4+ T cells are activated in response to antigen recognition and release cytokines such IL-2, IFN-γ, IL-12, and IL-6; (3) the cytokines recruit neutrophils, monocytes, CD8+ T cells, natural killer cells, and macrophages, which further boost the immune response [10,11]. Moreover, Takeuchi et al. demonstrated that Th-17 cells play a crucial role in BCG-driven anti-tumor immunotherapy by secreting IL-17, which recruits primordial neutrophils to the bladder tumor [12] (Figure 1).

Since Morales et al. first reported the treatment of bladder cancer through intravesical BCG therapy in 1976, numerous experimental studies and clinical trials have confirmed that intravesical BCG instillation is an effective immunotherapy for preventing postoperative recurrence [13]. The objective of in situ cancer treatment is to inhibit tumor development, improve survival chances, and extend the overall survival time [14,15]. BCG injections and transurethral resection of bladder tumor (TURBT) are the standard treatments for individuals with intermediate- to high-risk NMIBC, and they have been shown to effectively decrease tumor recurrence and enhance patient survival rates [16]. However, BCG is still inefficient in one-third of patients, leading to recurrence or progression to MIBC during or after treatment. The European Association of Urology (EAU) recommends induction BCG instillation for 6 weeks, followed by maintenance instillation every 3 years for high-risk NMIBC patients [6,17,18]. BCG induction and maintenance therapy decreased the likelihood of recurrence in bladder cancer patients by 32%, which was more effective compared to mitomycin C (*p* < 0.0001) [19]. VPM1002B is a genetically engineered variant of BCG, where the gene encoding Listeriolysin O (LLO), a hemolysin found in *Listeria monocytogenes*, has been inserted into the urease C gene to render it inactive [20]. After intravesical instillation of this variant, the pH in the bladder decreased due to the low level of urease C, and the activated LLO was able to reach the tumor cells after passing through the bladder mucosa. Furthermore, VPM1002B can be easily cleared by the host due to its higher immunogenicity compared to wild-type BCG, which enhances the therapeutic efficacy and reduces the side effects. In a clinical study involving VPM1002B, only 7 of 40 high-risk NMIBC patients showed disease progression, of which three cases progressed to MIBC and four developed metastasis, thereby confirming the higher efficacy of VPM1002B compared to wild-type BCG [21]. Despite its therapeutic success, however, BCG consists of live bacteria that can cause dysuria, hematuria, fever, bladder irritation, hepatitis, flu-like symptoms, etc. [22].

**Immune checkpoint inhibitors (ICIs):** Almost 20–30% of bladder cancer patients are diagnosed at the stage of MIBC, which is characterized by rapid progression, easy recurrence, and poor prognosis [23]. In addition, nearly half of the patients with MIBC who undergo radical cystectomy (RC) have distant metastases. MIBC is typically treated with RC following neoadjuvant chemotherapy (NAC), which has been shown to improve survival rates by reducing the chances of local recurrence and distant metastasis [24,25]. Neoadjuvant therapy is conducted before surgery to reduce the size of the tumor and lower the pathological stage, so as to extend patient survival and increase the success rate of radical resection [26,27]. However, about 50% of MIBC patients cannot tolerate chemotherapy due to adverse effects of the drugs and/or non-responsiveness [28].

Immune checkpoint inhibitors (ICIs) are a class of monoclonal antibodies that prevent checkpoint molecules such as cytotoxic T lymphocyte-associated protein 4 (CTLA-4) and programmed cell death receptor 1 (PD-1) from directly binding to their respective receptors [29,30]. T cell activation requires the recognition of antigenic peptides on the surface of antigen-presenting cells (APCs) by the T cell receptor (TCR) [31,32]. However, the binding of PD-1 on the surface of T cells to PD-L1 expressed on tumor cells or APCs can decrease T cell activation and limit cytokine production by relaying inhibitory signals. This mechanism is frequently used by the tumor cells to escape immune surveillance [33,34]. The ICIs block the binding of checkpoint receptors to the corresponding ligands, thus improving T cell activation and inhibiting immune escape [35]. PD-L1 overexpression on the bladder tumor cells is significantly correlated with advanced tumor grade and stage, as well as poor prognosis [36,37]. The United States’ Food and Drug Administration (FDA) has approved the use of PD-1 and PD-L1 inhibitors as the second-line treatment of metastatic or locally advanced bladder cancer and for the first-line management of PD-L1-positive patients who are ineligible for platinum-based chemotherapy [18,38]. Due to the efficacy of immunotherapy against bladder cancer and the low response rate of platinum NAC, the PD-1/PD-L1 inhibitors have been incorporated in the neoadjuvant immunotherapy for MIBC along with radiotherapy and chemotherapy [39]. At present, the FDA has approved two PD-1 inhibitors and three PD-L1 inhibitors for metastatic bladder cancer, namely, pembrolizumab, nivolumab, atezolizumab, durvalumab, and avelumab. The outcomes of various neoadjuvant immunotherapies against bladder cancer have been discussed in the following sections.

**Neoadjuvant immunotherapy:** Pembrolizumab has been approved by the FDA for treating metastatic urothelial bladder cancer in the case of platinum therapy failure and in patients unresponsive to cisplatin. The phase II prospective, single-arm PURE-01 (NCT02736266) trial was designed to determine the efficacy of pembrolizumab-based neoadjuvant immunotherapy in MIBC patients prior to RC treatment [40,41]. The pathologic complete response (pCR) rate was 37%, and the pathological descending response (pDR) rate was 55%. The most common adverse event (AE) was hyperthyroidism (23%), and 7% of the patients experienced an AE of grade 3 severity. In the KEYNOTE-057 trial, 41% of the patients with BCG-unresponsive bladder carcinoma in situ achieved complete response within 3 months of receiving pembrolizumab monotherapy, and 13% of the patients experienced grade 3 or 4 AEs, including arthralgia and hyponatremia [42]. The results of this trial indicate that pembrolizumab monotherapy can be a non-operative treatment option for people who do not conform to RC treatment. The multicenter, single-arm phase II trial ABACUS (NCT02662309) was conducted to evaluate the safety and efficacy of atezolizumab neoadjuvant treatment in 88 patients with MIBC [43,44]. The pCR rate was 31%, the 1-year recurrence-free survival (RFS) rate was 79%, and the incidence of grade 3–4 AEs was 11%. Furthermore, a phase II trial showed that atezolizumab monotherapy can benefit patients with BCG-unresponsive high-risk NMIBC; 27% of the 74 patients with carcinoma in situ achieved pCR, which lasted for 1 year in 48.9% of the patients. In addition, only 12 of the 129 patients developed MIBC or metastasis [45]. The pCR rates of MIBC patients after neoadjuvant immunotherapy were 37% and 31% in the PURE-01 and ABACUS trials, respectively, which was equivalent to that of standard NAC (37%) [46].

**Neoadjuvant double immunotherapy:** CTLA-4 inhibitors and PD-1 inhibitors are frequently used together for neoadjuvant immunotherapy. In the NABUCCO (NCT03387761) trial, the combination of the anti-CTLA4 antibody ipilimumab and the anti-PD-1 antibody nivolumab achieved a pCR of 46% and a pathological descending rate of 58% in bladder cancer patients. However, 41% of these patients experienced immune-related grade 3–4 AEs [47]. The CHECKMate-032 trial compared the safety and efficacy of combining ipilimumab with nivolumab with that of nivolumab monotherapy in patients with metastatic or advanced MIBC [48]. The median overall survival (mOS) of PD-L1^high^ patients in the combination treatment group was 15.3 months compared to only 9.9 months in the monotherapy group. Nevertheless, grade 3–4 AEs were more frequent in the combination group (39%) than in the monotherapy group (27%). For locally advanced urothelial carcinoma, inhibiting CTLA-4 and PD-1 before surgery may prove to be an effective preoperative therapeutic option.

**Neoadjuvant immunotherapy combined with chemotherapy:** Chemotherapy can alter the tumor microenvironment by increasing the infiltration of lymphocytes and reducing the infiltration of regulatory T cells. Moreover, chemotherapy can increase tumor antigen presentation via MHC-I [49,50] and promote immunogenic cell death. Due to this dual impact (cytotoxicity and immunological activation) of chemotherapy, the combination of ICIs and chemotherapeutic drugs may produce a synergistic anti-tumor effect.

The multicenter phase II BLASST-1 trial (NCT03294304) evaluated the safety and efficacy of nivolumab with gemcitabine plus cisplatin (GC) as a neoadjuvant treatment in patients with MIBC [51]. The pCR rate of 41 patients was 49%, and the pathological descending rate was 66%. Grade 3–4 AEs, such as renal failure and neutropenia, were recorded in 20% of the participants. The combination therapy was linked to significant pathological deterioration, with no additional toxicity or mortality. In addition, the results of the NCT02989584 trial showed that the combination of atezolizumab and GC decreased the pathological stage in 69% of the bladder cancer patients, and the adverse reactions of this combined therapy did not affect the outcomes of subsequent RC [52]. In the HCRN GU14-188 (NCT02365766) trial, pembrolizumab was used in conjunction with GC for MIBC patients. The pCR rate was 40%, the pathological degradation rate was 61%, and the frequency of grade 3–4 AEs due to blood toxicity was 44% [53]. In locally advanced urothelial cancer, NAC combined with pembrolizumab has controllable toxicity, and the pathological results have improved compared with previous controls.

In conclusion, preoperative NAC combined with immunotherapy in MIBC patients significantly reduced the tumor staging without increasing the frequency of AEs. Nevertheless, it is crucial to identify biomarkers to tailor personalized treatment plans and enable the selection of the appropriate monotherapy or combination therapy to achieve optimal efficacy.

**Table 1 vaccines-12-00150-t001:** Current status of immunotherapy for bladder cancer.

Immunotherapy for Bladder Cancer	Clinical Trial	Type of Therapy	Results
Immunotherapy of BCG vaccine	Intravesical BCG therapy (Morales et al., 1976 [13])	Biological immunotherapy	Effective in preventing postoperative recurrence
	BCG injections and TURBT	Standard treatment for intermediate- to high-risk NMIBC	Decreases tumor recurrence and enhances patient survival rates
	VPM1002B	Genetically engineered BCG variant	Better effectiveness compared to ordinary BCG. Only 7 of 40 high-risk NMIBC patients progressed, indicating improved efficacy
Immunotherapy of immune checkpoint inhibitors (ICIs)	Radical cystectomy (RC) with neoadjuvant chemotherapy (NAC)	Surgery + neoadjuvant chemotherapy	Effective in improving patient survival rates and preventing local recurrence and distant metastasis in MIBC [23,24]
	Immunocheckpoint inhibitors (ICIs) in neoadjuvant immunotherapy	Monotherapy with ICIs	Improved T cell activity, inhibited immune escape. Progress in reducing tumor size and stage in MIBC [38]
	Double immune combination in neoadjuvant immunotherapy	Combination of two ICIs	Synergistic effect in enhancing immune response. Further reduction in tumor size and stage in MIBC
	Immune combined with chemotherapy in neoadjuvant immunotherapy	ICIs + chemotherapy	Improved response rates compared to platinum-based NAC alone. Enhanced efficacy in neoadjuvant setting for MIBC
Neoadjuvant immunotherapy	PURE-01 (NCT02736266)—pembrolizumab	Neoadjuvant immunotherapy (single-arm, phase II)	Pathologic complete response (pCR) rate: 37%—Pathological descending response (pDR) rate: 55%—Most common adverse event (AE): hyperthyroidism (23%)—Grade 3 AE: 7%
	KEYNOTE-057—pembrolizumab for BCG-resistant in situ cancer (NCT02625961)	Pembrolizumab monotherapy	Complete response at 3 months for BCG-unresponsive bladder carcinoma in situ: 41%—Grade 3 or 4 AEs: 13% (including arthralgia and hyponatremia)
	ABACUS (NCT02662309)—atezolizumab	Neoadjuvant immunotherapy (phase II)	pCR rate after treatment: 31%—1-year recurrence-free survival (RFS) rate: 79%—Incidence of grade 3–4 AEs: 11%
	Atezolizumab for non-BCG-reactive, high-risk NMIBC (NCT02108652)	Atezolizumab treatment	pCR rate for carcinoma in situ: 27%—1-year pCR lasting rate: 48.9%—Only 12 out of 129 patients developed MIBC or metastatic diseases
Neoadjuvant double immunotherapy	NABUCCO (NCT03387761)	Double immunotherapy with ipilimumab and nivolumab	Pathologic complete response (pCR): 46%—Pathological descending rate: 58%—Grade 3–4 immune-related adverse events: 41%
	CHECKMate-032 (NCT01928394)	Combination of ipilimumab with nivolumab vs. nivolumab alone	Combo group (high PD-L1 expression): median overall survival (mOS) of 15.3 months—Monotherapy group: mOS of 9.9 months—Grade 3–4 adverse events: combo (39%) vs. monotherapy (27%)
Neoadjuvant immunotherapy combined with chemotherapy	BLASST-1 (NCT03294304)	Neoadjuvant immunotherapy (nivolumab) with gemcitabine plus cisplatin (GC)	Pathologic complete response (pCR): 49%—Pathological descending rate: 66%—Grade 3–4 adverse events: renal failure and neutropenia in 20% of participants
	NCT02989584	Neoadjuvant immunotherapy (atezolizumab) with gemcitabine plus cisplatin (GC)	Pathological descending phase: 69%—Adverse reactions did not affect subsequent radical cystectomy (RC)
	HCRN GU14-188 (NCT02365766)	Neoadjuvant immunotherapy (pembrolizumab) with gemcitabine plus cisplatin (GC)	pCR: 40%—Pathological descending rate: 61%—Incidence of grade 3–4 adverse events (blood-related): 44%

## 3. Effect of Pelvic Lymph Node Dissection on Immunotherapy for Bladder Cancer

Approximately 30% of bladder cancer cases, which affect over 400,000 people annually, are muscle-invasive [54]. For recurring high-risk NMIBC and MIBC, RC in conjunction with bilateral pelvic lymph node dissection (PLND) is the most effective treatment [55,56]. However, the sensitivity of preoperative cross-sectional imaging for detection of positive pelvic lymph nodes is only 52%, which can lead to substantial under staging [57,58,59]. Therefore, a thorough bilateral PLND increases the accuracy of surgical staging and may provide a survival benefit for patients with MIBC regardless of lymph node involvement [60,61] (Table 2).

**Lymphatic drainage in the bladder:** The obturator, presacral, and internal and external iliac lymph nodes are the primary lymphatic drainage sites in the bladder, and the para-aortic, interaortocaval, paracaval, and common iliac nodes are supplied by secondary lymphatic drainage [62,63,64]. Based on the typical metastatic spread of primary MIBC [62,65,66,67], numerous studies have investigated the therapeutic significance of bilateral PLND in patients diagnosed with bladder cancer.

Abol-Enien et al. [62] found that approximately 39% of bladder cancer patients had bilateral lymph node metastases based on the assessment of tissues obtained by PLND. In addition, surgical probing after technetium nanocolloid injection and single-photon emission computed tomography (SPECT) demonstrated that 52% of lymph nodes were situated outside of the reconstructed pelvic cavity [62,63,65]. Leissner et al. identified 12 sites with varying probabilities of metastatic growth in bladder cancer patients, and the obturator lymph nodes were most frequently afflicted [68]. Furthermore, the patients were more likely to develop metastases in the interaortocaval and precaval areas (2.9%) and above the common iliac bifurcation (6.9%). Other studies have shown that up to 41% of positive lymph nodes are located in the region above the common iliac artery [69]. Tarin et al. [70] examined 591 patients who underwent RC and detected lymph node involvement in 114 patients. The percentage of patients with lymph node involvement ranged from 6% to 40% depending on the pT2, pT3, and pT4 stages. In addition, seven patients with skip lesions had negative lymph node involvement in the pelvis. Given the rarity of skip lesions, this finding could be due to mislabeling or the failure to detect positive lymph node in the anatomically true pelvis.

Therefore, numerous hypotheses involving primary lymph node drainage are supported by available data. Even in the case of unilateral tumors, there seems to be evidence against unilateral PLND due to the bilateral lymphatic drainage of the bladder. Secondly, the pelvic region is the principal area of lymphatic outflow and potential metastatic seeding. A modest but not insignificant percentage of metastatic deposits lie outside the pelvic borders, indicating that the underlying drainage is more extensive than the perivesical lymphatics. Due to the possibility of extra pelvic metastatic deposits, lymph node dissection may have to extend beyond the pelvic region.

**Templates of PLND:** The anatomical extent of PLND is classified as standard, super-extended, extended, and limited according to the EAU Working Group on MIBC. Only the bilateral obturator fossa is dissected during limited PLND [71,72,73] (Figure 2a). Standard PLND includes the medial and lateral walls of the bladder, as well as the lymph nodes between the lower division of the inguinal ligament and the genitofemoral nerve, the upper division of the common iliac artery, and the inguinal ligament [72]. This PLND template typically involves the distal common iliac, obturator, external iliac, and hypogastric nodes on either side [73,74] (Figure 2b). Extended PLND refers to the excision of nodes from the area that includes the aortic bifurcation, internal iliac arteries, circumflex iliac vein, common iliac vessels, and genitofemoral nerve [75] (Figure 2c). Super-extended PLND further proceeds proximally to the inferior mesenteric artery root [71,76] (Figure 2d).

**Table 2 vaccines-12-00150-t002:** Effect of pelvic lymph node dissection on immunotherapy for bladder cancer.

Immunotherapy for Bladder Cancer	Clinical Trial	Type of Therapy	Results
Lymphatic drainage of the bladder	Lymph node dissection studies	Surgical procedure	Abol-Enien et al. [61]: Found that approximately 39% of patients with prostate cancer had lymph nodes on both sides, and bilateral lymph node metastases were common. Up to 52% of lymph nodes were situated outside the reconstructed pelvic cavity
			Leissner et al. [68]: Identified 12 anatomical sites with varying probabilities of metastatic deposits, with the obturator groups being the most frequently afflicted. Higher probabilities of metastases were observed in the interaortocaval and precaval areas (2.9%) and above the common iliac bifurcation (6.9%)
			Tarin et al. [69]: Studied 591 individuals with radical cystectomy, finding LN involvement in 114 cases. The percentage of patients with LN involvement ranged from 6% to 40%, depending on tumor stage (pT2, pT3, and pT4). Identified cases with negative lymph node testing in the pelvis, possibly due to skip lesions or mislabeling
Templates of pelvic lymph node dissection	EAU Working Group	Limited PLND	Dissects only bilateral obturator fossa
		Standard PLND	Dissects medial and lateral walls of the bladder, lymph nodes between the lower division of the inguinal ligament and the genitofemoral nerve, upper division of the common iliac artery, and the inguinal ligament. Includes distal common iliac, obturator, external iliac, and hypogastric nodes on either side
		Extended PLND	Excises nodes from the area that includes the aortic bifurcation, internal iliac arteries, circumflex iliac vein, common iliac vessels, and genitofemoral nerve
		Super-extended PLND	Dissection proceeds proximally to the inferior mesenteric artery root

**Significance of PLND in RC:** A number of studies have demonstrated that PLND can increase long-term survival probability and lower the risk of postoperative lymph node metastases in patients undergoing RC [77,78,79,80,81,82,83]. An epidemiological study based on the Surveillance, Epidemiology, and End Results (SEER) database showed that patients with fewer than three excised lymph nodes had a greater risk of dying than those with higher lymph node removal (HR 0.41 to 0.58) [81]. Larcher et al. [84] analyzed 1376 bladder cancer patients from the SEER database and found that patients who underwent pelvic lymphadenectomy had significantly higher 5-year survival rates than the non-PLND group. Bruins et al. [85] conducted a retrospective case study on 637 bladder cancer patients and found that RC combined with PLND could improve the survival rate and reduce the possibility of tumor recurrence, especially for those with pathological stages of cT1 and cTis. In contrast, the impact of PLND was limited for patients with cTa stage tumors.

Despite the improvement in outcomes, there is currently no consensus on the relevance of PLND in patients with a lack of lymph node involvement, mainly due to the prolonged procedure and the morbidities related to sympathetic nerve injury, such as lymphocele, lymphedema, and incontinence [86]. Therefore, adequate risk assessment and postoperative rehabilitation guidance should be provided to reduce the discomfort of patients and ensure their quality of life. However, the risks of RC alone vastly outweigh that when PLND is performed concomitantly. Moreover, there is evidence that lymphadenectomy is also beneficial for individuals with node-negative illness ontologically [87,88]. In fact, increasing lymph node yield in individuals with negative nodes appears to enhance cancer outcomes.

In conclusion, PLND is associated with prognostic and therapeutic advantages in bladder cancer cases involving cystectomies. The greater risk connected with PLND is outweighed by these oncological benefits.

**Effect of PLND on immunotherapy:** For MIBC patients who can receive cisplatin, the recommended course of treatment is RC with bilateral PLND, followed by chemotherapy and neoadjuvant therapy [89]. However, NAC is only accessible to 20% of the eligible patients and has not yet gained widespread use as a treatment for MIBC. Clinical studies have shown that a little over half of MIBC patients have pre-existing contraindications that make them ineligible for cisplatin treatment, and several patients decline therapy altogether [90,91]. In Europe, RC is typically performed 6–8 weeks after the diagnosis of MIBC, and delaying surgery by more than 12 weeks can increase the risk of death [92]. This time frame offers a distinctive chance to assess the drugs for neoadjuvant treatment. ICIs have significantly altered the landscape of bladder cancer treatment. A novel approach to neoadjuvant therapy for NMIBC may involve incorporating short-course pre-RC immunotherapy [93]. However, according to National Comprehensive Cancer Network recommendations, bilateral PLND should be performed for MIBC patients scheduled for radical or partial cystectomy, with the removal of only a small amount of the obturator, internal, and common iliac lymph nodes. A more extensive PLND involving the paraaortic and paracaval lymph nodes can be performed if possible [18]. However, PLND may damage the immune system by disrupting the integrity of the lymphatic system. Lymph nodes are the primary sites of antigen presentation, T cell activation, and immune memory generation, and they are therefore crucial to the anti-tumor immune response [94,95,96]. However, it is necessary to remove the lymph nodes proximal to the tumors during surgical resection in order to clear any residual metastatic cells [97,98,99,100]. On the other hand, destroying the integrity of the lymphatic system may also lead to the accumulation of lymph fluid and complications such as lymphatic embolism, which further the immune function [101].

A recent clinical study showed that preserving lymph node integrity before immunotherapy may help improve therapeutic efficacy against solid tumors [102,103]. Thus, lymph node dissection may not always be advantageous for patients. Therefore, once tumor diagnosis is confirmed, various treatment methods and their order and time points should be carefully considered in order to avoid interference and conflict between treatment methods and to improve the outcomes.

**Effect of PLND on postoperative immunotherapy:** The impact of PLND on postoperative immunotherapy for bladder cancer is a subject of considerable academic interest. PLND, ranging from limited to super-extended dissections, plays a crucial role in the surgical management of bladder cancer by evaluating the extent of lymphatic involvement. Though the immediate focus of PLND is tumor staging and guiding therapeutic decisions, its potential influence on subsequent immunotherapy outcomes warrants further investigation. Understanding the anatomical coverage of PLND templates is crucial in order to elucidate the systemic immunological responses postoperatively. The interaction between the resected lymph nodes and the intricate immune microenvironment may influence the efficacy of immunotherapeutic interventions. Future studies should explore the nuanced interplay between PLND and immunotherapy in bladder cancer and consider factors such as lymph node status and template extent in order to optimize postoperative treatment strategies and enhance patient outcomes.

**Effect of PLND on adjuvant immunotherapy:** The intricate relationship between PLND and adjuvant immunotherapy in bladder cancer is a key area of research for urological tumors. PLND allows for comprehensive lymph node evaluation, which impacts both staging and therapeutic decisions. Furthermore, the extent and nature of PLND are key determinants of the outcomes of adjuvant immunotherapy in bladder cancer patients. The resected lymph nodes are potential reservoirs of residual disease and immune modulation, and they can therefore significantly shape the postoperative immunological landscape. The immunotherapeutic response is intricately linked to the adequacy of PLND, as it dictates the extent of lymphatic involvement and potential immune stimulatory or inhibitory signals. The nuanced interplay between PLND and adjuvant immunotherapy also warrants a thorough investigation, with focus on optimizing PLND templates to enhance the efficacy of immunotherapeutic interventions. Insights derived from these studies hold the promise of refining postoperative adjuvant strategies, ensuring a synergistic approach that harnesses both surgical and immunotherapeutic modalities for improved outcomes in bladder cancer patients.

## 4. Conclusions and Discussion

The landscape of bladder cancer treatment has witnessed significant advances with the advent of immunotherapeutic approaches, particularly the BCG vaccine and ICIs. Despite their demonstrated efficacy, the outcomes of immunotherapies are limited by drug resistance, immune toxicity, and treatment failure [104,105]. The future trajectory of immunotherapy in bladder cancer necessitates a multi-faceted approach. Combination therapies, the development of novel drugs, and personalized treatments are emerging as key avenues for further exploration. Ongoing studies on the role of immune cells, cytokines, and chemokines in bladder tumor invasion, proliferation, and metastasis can provide novel insights for devising alternative therapies, particularly for patients who are averse to or ineligible for cystectomy.

In the era of radiation and chemotherapy, aggressive lymph node dissection remains a recommended practice, since it contributes to enhanced local control and precise tumor staging. However, in the evolving landscape of immunotherapy, a delicate balance is required between surgery and multidisciplinary treatments, as well as between local and systemic therapeutic modalities, in order to optimize treatment outcomes.

Future research should focus on the intricacies of immune responses in bladder cancer and devising strategies to overcome challenges associated with immunotherapy. The ultimate goal is to offer a spectrum of effective and personalized treatments that not only address the complexities of the disease but also align with individual patient needs and preferences. In navigating this dynamic landscape, the integration of immunotherapy into the broader treatment paradigm underscores the importance of a holistic and patient-centered approach to bladder cancer management.

## Figures and Tables

**Figure 1 vaccines-12-00150-f001:**
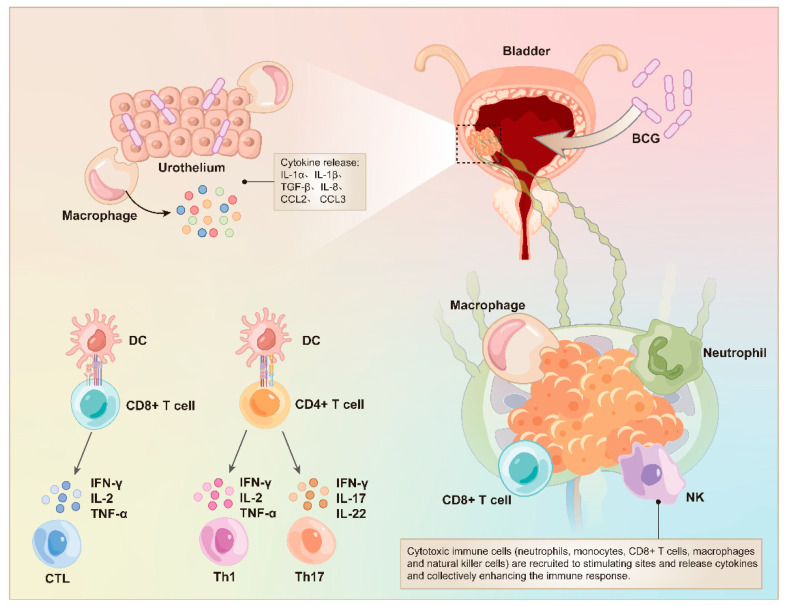
Immune response induced by live BCG infusion. BCG attaches to the urothelium and is phagocytosed by the macrophages and dendritic cells. Following internalization, BCG antigens are presented to CD4 T cells and CD8 T cells by the dendritic cells, resulting in the activation of Th1 cells, Th17 cells, and cytotoxic T lymphocytes, which form the adaptive immune response. Abbreviations: BCG, Bacillus Calmette–Guérin; DC, dendritic cells; NK, natural killer; CTL, cytotoxic T lymphocytes; TNF, tumor necrosis factor; IFN, interferon; IL, interleukin; TGF, transforming growth factor; CCL, C-C motif chemokine ligand.

**Figure 2 vaccines-12-00150-f002:**
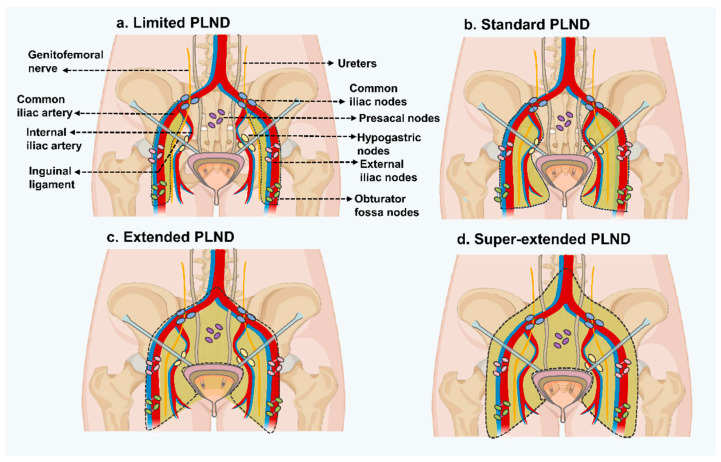
Schematics of limited, standard, extended, and hyperextended templates for PLND.

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
