# Peer review of "Bladder Cancer: Immunotherapy and Pelvic Lymph Node Dissection"

_vaccines, 2024, doi:10.3390/vaccines12020150_

Round 1

Reviewer 1 Report

Comments and Suggestions for Authors

This is an interesting and comprehensive review. However, the english language use  is simply not acceptable, and a massive re-write with help from a consulting service company or an English speaking colleague is necessary.

Comments on the Quality of English Language

This is an interesting and comprehensive review. However, the english language use  is simply not acceptable, and a massive re-wrtie with help from a consulting service company or an English speaking colleague is necessary.

Author Response

Thank the reviewer for your recognition of our review. In terms of language expression, we admit that it is indeed inadequate, we have carried out the overall polish and improvement of the English of the full text, looking forward to your recognition and affirmation, thank you.

Reviewer 2 Report

Comments and Suggestions for Authors

The review covers important aspects of bladder cancer, but the information is somewhat scattered. I suggest elaborating a table indicating the clinical trial, the type of therapy (e.g. immunotherapy with check point inhibitors, immunotherapy with neoadjuvant, immunotherapy with double neoadjuvant, immunotherapy combined with chemotherapy, etc.), with the relevant results of each one of them. And in another area of the table include immunotherapy with dissection, or with pelvic lymph node dissection, etc.); If you could indicate the clinical phase, I think that way we would have an overview of the most important results of each therapeutic strategy.

I also believe that the discussion and conclusions remain short, and it would be important to include a more in-depth review of ongoing clinical trials (www.clinicaltrials.gov).

minor observations:
In line 87 the reference number is missing.

Author Response

First of all, thank the reviewer for your comments. According to your suggestion, we have added two tables in the paper, which may help us better summarize the scattered information in the paper. In addition, we have rewritten the discussion and conclusion part of the paper, and we have further deepened our discussion and conclusion. We have added the reference in the line 87.

Reviewer 3 Report

Comments and Suggestions for Authors

I read with great enthusiasm the manuscript by Fan et.al. because it seems to attempt to address an important clinical question - what is the implication of LND in the area of immuotherapy for bladder cancer? This is a really important question, especially with preclinial data suggesting intact lymph nodes may be necessary for ICI to work effectively (the original paper was cited in the manuscript). A little to my disappointment, authors spent almost all parts of the manuscript just discussing the role of immunotherapy and LND in bladder cancer as separate entities. There are already lots of reviews on these topics and we really don't need another one to re-elaborate them. Eventually in the last part of the manuscript, authors finally start to discuss what is supposed to be the most interesting topic but the discussion was very superficial. They only mentioned studies of ICI in the neo-adjuvant setting and failed to include studies in the postop/adjuvant setting - this is really where the money is. I strongly advise the authors to revise this last part so that this review can bring new information to the field. Again, the literature really does not need another review to only discuss either ICI or LND in bladder cancer. 

Minor comment - when discussion LND in bladder cancer, the author should include results from the recent phase 3 SWOG S1011 trial where standard vs extended LND in radical cystectomy was compared. 

Author Response

Thank the reviewer for your comments. In this review, we mainly focus on the important influence and role of LND in bladder cancer immunotherapy, but we do have some incomplete places. Thank you for your good advice to us. In the last part of the article, we discussed the postoperative impact of LND on bladder cancer immunotherapy and the help of adjuvant therapy for bladder cancer immunotherapy. The discussions in these two parts really add to the depth and breadth of our review. In addition, thanks to the reviewer's suggestion, we have added the recent phase 3 SWOG S1011 trial to the reference.

Round 2

Reviewer 3 Report

Comments and Suggestions for Authors

The manuscript was revised appropriately. No additonal comments. 

Author Response

Thank you for your recognition of our revision!